# Recent Advances in Synthesis of Graphite from Agricultural Bio-Waste Material: A Review

**DOI:** 10.3390/ma16093601

**Published:** 2023-05-08

**Authors:** Yee Wen Yap, Norsuria Mahmed, Mohd Natashah Norizan, Shayfull Zamree Abd Rahim, Midhat Nabil Ahmad Salimi, Kamrosni Abdul Razak, Ili Salwani Mohamad, Mohd Mustafa Al-Bakri Abdullah, Mohd Yusry Mohamad Yunus

**Affiliations:** 1Faculty of Chemical Engineering & Technology, Universiti Malaysia Perlis (UniMAP), Arau 02600, Malaysia; yapyeewen1129@gmail.com (Y.W.Y.); nabil@unimap.edu.my (M.N.A.S.); kamrosni@unimap.edu.my (K.A.R.); mustafa_albakri@unimap.edu.my (M.M.A.-B.A.); 2Geopolymer and Green Technology, Centre of Excellence (CEGeoGTech), Universiti Malaysia Perlis (UniMAP), Arau 02600, Malaysia; shayfull@unimap.edu.my (S.Z.A.R.);; 3Faculty of Electronic Engineering & Technology, Universiti Malaysia Perlis (UniMAP), Arau 02600, Malaysia; 4Faculty of Mechanical Engineering & Technology, Universiti Malaysia Perlis (UniMAP), Arau 02600, Malaysia; 5Terkemuka Solution Sdn Bhd, Seksyen U10 Alam Budiman, Shah Alam 40170, Malaysia; ujangcordy@gmail.com

**Keywords:** graphite, agricultural, bio-waste material, carbon, graphitization

## Abstract

Graphitic carbon is a valuable material that can be utilized in many fields, such as electronics, energy storage and wastewater filtration. Due to the high demand for commercial graphite, an alternative raw material with lower costs that is environmentally friendly has been explored. Amongst these, an agricultural bio-waste material has become an option due to its highly bioactive properties, such as bioavailability, antioxidant, antimicrobial, in vitro and anti-inflammatory properties. In addition, biomass wastes usually have high organic carbon content, which has been discovered by many researchers as an alternative carbon material to produce graphite. However, there are several challenges associated with the graphite production process from biomass waste materials, such as impurities, the processing conditions and production costs. Agricultural bio-waste materials typically contain many volatiles and impurities, which can interfere with the synthesis process and reduce the quality of the graphitic carbon produced. Moreover, the processing conditions required for the synthesis of graphitic carbon from agricultural biomass waste materials are quite challenging to optimize. The temperature, pressure, catalyst used and other parameters must be carefully controlled to ensure that the desired product is obtained. Nevertheless, the use of agricultural biomass waste materials as a raw material for graphitic carbon synthesis can reduce the production costs. Improving the overall cost-effectiveness of this approach depends on many factors, including the availability and cost of the feedstock, the processing costs and the market demand for the final product. Therefore, in this review, the importance of biomass waste utilization is discussed. Various methods of synthesizing graphitic carbon are also reviewed. The discussion ranges from the conversion of biomass waste into carbon-rich feedstocks with different recent advances to the method of synthesis of graphitic carbon. The importance of utilizing agricultural biomass waste and the types of potential biomass waste carbon precursors and their pre-treatment methods are also reviewed. Finally, the gaps found in the previous research are proposed as a future research suggestion. Overall, the synthesis of graphite from agricultural bio-waste materials is a promising area of research, but more work is needed to address the challenges associated with this process and to demonstrate its viability at scale.

## 1. Introduction

Carbon is one of the most abundant chemical elements. It is the sixth element in the periodic table and is able to be bonded with four valence electrons. Due to this reason, carbon is unique as it is able to form stable chemical bonds with many other elements. Carbon is also known as the key element of life as it is the building block for most of the living beings on Earth [1,2]. Many of the most important molecules in the human and animal body, including proteins, deoxyribonucleic acid (DNA), ribonucleic acid (RNA), sugars and fats, are derived from the carbon backbone. Carbon can exist purely or can form a compound with other elements. Pure carbon can exist in different forms, which are also known as allotropes of carbon. For example, allotropes of carbon include diamond, graphite, amorphous carbon, lonssdaleite, carbon nanotubes and fullerene [3,4,5].

As mentioned, a carbon atom in the excited state has four valence electrons, and it is able to form various allotropes. There is allotropy when an element exists in more than one type of crystalline form. Allotropes have identical chemical properties; however, they have different physical properties, as the atoms of the element are bonded together in various structures [6]. For example, diamond is well known as a very hard and inert material [7]. This is due to the fact that there is a strong covalent bond between the carbon atoms, bonded together to form a cubic lattice of 3D tetrahedra [8]. At the same time, the carbon atoms in graphite are bonded together in sheets of hexagonal lattices—bonding between layers is via relatively weak van der Waals bonds, which allow the layers to be easily separated and to glide past each other [9].

Graphite is the crystalline form of carbon, which is formed by the arrangement of carbon atoms in a hexagonal structure. The structure of graphite is illustrated in Figure 1. In graphite, the carbon atom arranged in the honeycomb lattice has an average covalent bond length of 0.14 nm [10]. Graphite is the most stable form of carbon, which naturally exists under standard conditions [6]. Moreover, graphite is also made up of numerous graphene layers. The interplanar spacing, *d*, of the (002) graphite is approximately 0.34 nm [11]. When single-atom graphene layers are stacked and held together by Van der Waals weak forces, graphite is formed [12]. Due to the weak Van der Waals forces located in between the graphene layers, graphite develops a soft and slippery property as the Van der Waals force can be easily substituted by an external force.

Natural graphite can be obtained by extracting it from graphite mines. Besides mining, graphite can also be manmade by synthesizing from carbon containing compounds [13]. Synthetic graphite is essentially produced by the two subsequent processes, which are the carbonization process of carbon precursors and the graphitization process of the amorphous carbon [14,15,16,17]. Both carbonization and graphitization are common heat treatment processes. Carbonization is a process to produce carbon or charcoal. Through the carbonization process, carbon can be produced by pyrolysis. During the process, the carbon precursor will be heated at a high temperature under an inert condition. This process helps to expel the absorbed volatiles in the precursors and, thus, produces only porous carbon or char [18,19]. Meanwhile, graphitization is a heat treatment process that heats a carbon precursor at high temperatures to transform the amorphous carbon into crystalline graphite [20]. For synthetic graphite production, natural carbon resources such as coal (bituminous coal [21]), anthracite [22], fossil fuels (lignite [23]) and carbonized and graphitized lignocellulosic materials (Kesambi wood [24], palm waste [25,26], etc.) have been used.

A precursor is a biochemical compound that forms another compound after undergoing a chemical reaction [27]. A carbon precursor is a raw material that is used in producing carbon fiber after a heat treatment process. Since carbon is the main element for most of the world’s living organisms, many resources can be used as carbon precursors. However, compounds with abundant carbon content are more desirable since they can produce higher amounts of carbon. There are three carbon precursors that are commonly used to produce carbon fiber. The first precursor is cellulose, which was introduced in 1871. The second carbon precursor is pitch and polyacrylonitrile (PAN), while the third precursor is lignin [28]. Among them, two are natural precursors (cellulose and lignin), while PAN is a synthetic precursor. In this review paper, the discussion will focus on the synthesis of graphite from agricultural biomass waste, which is from a natural carbon precursor that contains both cellulose and lignin [29].

Agricultural biomass waste is worth discussing in terms of environmental problems. Although they are considered as waste, their potential bioactive properties, such as bioavailability, antioxidant [30], antimicrobial [31,32], in vitro and anti-inflammatory properties, could benefit human life. As mentioned in [33], the by-products from the tomato processing industry, such as lycopene, beta-carotene, glutamic acid or aspartic acid, can be revalorized and incorporated as nutrients in functional foods. The carotenoids’ bioavailability in tomato by-products is a key factor that helps in the prevention of different chronic diseases and carcinogenesis [33]. According to Baysal et al. [32], olive leaves are proven to have antioxidant and antimicrobial functions. By varying the drying method of olive leaves from different sources, one can obtain a direct effect on its phenolic compound content, which affects its functionality. Another research work used hydrolysates of fruit or vegetable waste in the cultivation of oleaginous red yeast that produced carotenoids. The efficacy of the yeast carotenoid extract was demonstrated on aggressive breast cancer cell lines, indicating its potential as a replacement for expensive synthetic carotenoids or plant-based/algal pigments [34]. Based on the previous discussion, there are many benefits of agricultural waste in various applications. Therefore, in this review article, the usage of agricultural biomass waste with high carbon content in the synthesis of graphite will be evaluated.

Graphite can be produced by both synthetic and natural carbon precursors. Due to the depletion of natural resources [35], this paper will focus on the previous research that has been conducted on the synthesis of crystalline graphite by using potential agricultural biomass waste materials as the organic natural carbon precursor. We also consider the importance of utilizing agricultural biomass waste and the types of potential biomass waste used as carbon precursors. The pre-treatment method is one of the crucial steps to optimize the conversion of biomass waste material into carbon precursors. Thus, the pre-treatment methods will also be reviewed. In addition, this paper also discusses various synthesis methods for graphite. Finally, the gaps discovered in the previous literature will be suggested as a future research topic.

## 2. Carbon Precursors from Agricultural Bio-Waste Materials

The graphite synthesis process has become increasingly prevalent in recent years. Carbon is the basic element and is arranged in a crystalline structure, which forms graphite [10]. Thus, carbon is selected as a precursor for graphite production. Numerous studies have suggested the application of biological waste as a potential carbon precursor for graphite synthesis [36,37,38,39,40,41]. Agricultural sectors have generated a large amount of bio-waste over the years. This bio-waste, also known as an agricultural biomass material, is an undesired waste product that is discharged from various agricultural activities [42]. As the amount of waste is increasing, the utilization of agricultural bio-waste has become a necessity. Biomass waste, especially that with high carbon content, has been used as a feedstock for graphite since it is ubiquitous, environmentally friendly [43], renewable and cheap [44].

Besides graphite, the process that is used to convert biomass waste materials into products with concentrated carbon content is also applied in the biochar industry. Biochar is a carbon-rich product that is produced by the process of pyrolysis or gasification [45].

According to Xie et al. [46], the long chains of carbon, hydrogen and oxygen compounds are commonly found in most biomass materials. The carbon content in biomass materials can be as high as 55 wt%. This high carbon content qualifies them to be applied as carbon precursors to produce graphitic carbon. There are different types of biomass materials with high carbon content, including palm kernel shells [47,48], waste wood [49], various types of nut shells (cashew nut shell, pecan nut shell, macadamia nut shell) [50,51,52] etc. To convert these agricultural bio-waste materials into high-carbon products, a pre-treatment process needs to be conducted prior to the pyrolysis process. Pre-treatment is important to ensure that the material is cleaned, dried and ready for the carbonization process. Thus, in the next section, the importance of the utilization of agricultural bio-waste materials, their types and their pre-treatment methods will be discussed.

### 2.1. Importance of Utilizing Agricultural Bio-Waste Material

The United Nations Environment Programme’s International Resource Panel has released a new report that provides empirical evidence of the levels of natural resources expended by humans [35]. The past trends of natural resource consumption can help to mirror the future use trends of natural resources [35]. It is supported by work conducted by the World Bank, who stated that the world’s consumption of resources will triple by 2050 [53]. According to [54], global commodities such as biomass, fossil fuels and metals are predicted to double over the next forty years, while yearly waste disposal is estimated to rise by 70% by 2050. Therefore, a substitute for energy and natural resources is indeed necessary for future purposes.

Across the world, the amount of agricultural biomass waste that can be produced in a year is 998 million tonnes [55]. Agricultural biomass wastes or residues are primarily plant stems, crop stalks, leaves, roots, fruit peels and seed or nut shells that are typically discarded or incinerated, but they can be a potentially valuable source of feed material in practice [56]. There are a number of industrial activities that severely exacerbate the amount of agricultural waste. For example, Brazil is one of the largest producers of heart of palm, generating tons of agricultural bio-waste from the palm tree due to the extraction and processes performed while generating the product [57]. The production of agricultural bio-waste can be seen in daily industrial activities. For example, the fruit juice industry produces a sizable number of agricultural bio-wastes, such as orange peels, sugarcane fiber and different types of fruit stones [58,59]. The amount of agricultural waste has accumulated and is estimated to grow year by year. Therefore, the disposal and management of agricultural waste has been a great challenge. Presently, in the European Council Strategy for the Adriatic and Ionian Region (EUSAIR) area, most of the agricultural waste management methods that are commonly practiced by communities are contributing to environmental problems. The various disposal methods including open burning, landfilling, dumping and random piling, creating environmental pollution [60]. Meanwhile, in Malaysia, it was recorded that 1.2 million tonnes of agricultural waste are discarded by landfilling annually [61]. It is clear that the utilization of agricultural bio-waste is crucial in order to avoid further devastation to the Earth.

Owing to the above problems, a method to resolve the depletion of natural resources in the coming era, the issues brought by the rise in the amount of agricultural bio-waste and the improper management methods for agricultural bio-waste must be addressed and implemented. Due to these concerns, various efforts are being made to exploit alternative energy sources by converting biomass waste into new energy. In the 1970s, agricultural bio-waste materials were treated as one of the most important sources of fuel [62]. However, they lost their commercial value and were replaced by oil due the sudden drop in oil prices in 1986 [62]. Thus, this provides evidence that agricultural bio-waste materials can be used as feedstock sources for biofuel production [63]. Moreover, the high carbon content of biomass materials means that they can be converted into high-energy biochar via thermochemical treatment [64] and carbon source materials after undergoing the pyrolysis process [65]. According to Sun et al. [66], bio-waste management cost–benefit analyses are expected to become a driving force in selecting a bio-waste disposal method. By converting agricultural bio-waste material into new functional materials, we can utilize the unwanted waste and add new commercial value to it. Therefore, we can address the concern regarding high-cost waste management methods that drive environmental pollution.

Utilizing biomass waste to produce carbon materials offers a promising strategy for sustainable waste management and the advancement of renewable energy technologies. The produced carbon from biomass waste materials not only contributes to waste management, but has wide applications in various fields. These include energy storage devices such as batteries [67] and capacitors [68], water purification systems [69,70] and catalyst supports in chemical reactions [71].

### 2.2. Types of Agricultural Bio-Waste Material as a Carbonaceous Precursor

Carbonaceous material is an organic substance that has high carbon content. Carbonaceous materials can serve as carbon precursors for the development of various materials with high application potential, such as activated carbon (AC), ordered mesoporous carbon (OMC), carbon nanotubes (CNTs), fullerene (C60) and graphene (GE) [72]. In this article, the types of agricultural bio-waste materials that are commonly used as carbonaceous precursors will be reviewed.

Agricultural wastes are lignocellulosic materials that contain cellulose, hemicellulose and lignin as basic structural components [29], with general content of 9–80%, 10–50% and 5–35%, respectively [73]. All three components, hemicellulose, lignin and cellulose, are high in carbon content as their structure is built from a carbon backbone. However, lignin has the highest content of carbon and lower oxygen content [74]. Lignin is an organic polymer; generally, it is present in all vascular vegetables as it is an important component in the formation of the cell wall in plants. Lignin’s carbon content is approximately 60% [75].

As reported by Rybarczyk et al. [76], the general composition of rice husks includes cellulose (38%), hemicellulose (18%), lignin (22%) and silica (SiO_2_), and carbon is successfully extracted from them. Thus, it is suggested that rice husk, as an agricultural waste material, can become a carbonaceous precursor, which then becomes a resource for carbon production.

Destyorini et al. [77] conducted research on the transformation of the amorphous carbon structure from agricultural bio-waste materials into a graphitic nanostructure. The article reported the use of coconut coir as a carbon precursor in the production of graphitic carbon with a simple and low-energy process. A well-defined lattice fringe of a graphitic nanostructure that corresponds to pure graphite was revealed using high-resolution transmission electron microscopy (HRTEM). It was proven that the production of high-quality graphite can be achieved by using agricultural bio-waste, namely coconut coir, as a low-budget carbon precursor.

According to the evaluations in [78,79,80,81,82,83,84,85], the trend of using agricultural bio-waste as a carbonaceous precursor is widely demonstrated. Various agricultural bio-waste materials can be further converted into new energy resources or useful materials, which helps in adding economic value to these waste materials. Examples of the new materials formed are biochar, activated carbon, graphitic carbon, etc. However, a cost–benefit analysis for the conversion of waste materials into new materials has not been reported. This gives rise to significant concern: the cost-effectiveness of the waste conversion process is of great importance, as an uneconomic process will further burden the industry in the management of agricultural bio-waste. Moreover, the process of carbonization to produce a carbon precursor deserves attention as it is a high-complexity processing method that could indirectly magnify the waste management issue.

Table 1 shows the proximate and ultimate analysis of agricultural bio-waste that is commonly used as a carbonaceous precursor. The table shows that for the proximate analysis, the volatile content has the largest weight proportion in agricultural waste. The ultimate analysis, which determines carbon (C), hydrogen (H), nitrogen (N), oxygen (O) and sulfur (S), shows that all lignocellulosic agricultural wastes have high carbon content, ranging from 37.8% to 57.5% by weight. This proves that lignocellulosic biomass wastes are an ideal carbon precursor, as they already have high carbon content before the carbonization process.

### 2.3. Pre-Treatment Method for Preparation of Carbon Precursor

To convert agricultural bio-waste materials into a carbon precursor, numerous pre-treatment methods needed to be conducted before undergoing the pyrolysis or heat treatment process. The agricultural bio-waste feedstock must be in a well-dried condition. As introduced by Rout et al. [86], the moisture content of the material needs to be controlled below 5%. This is because when the moisture content of the feedstock is below 5%, it can help to enhance the pyrolysis products’ quality as less energy is needed to further dry out the extra moisture. The preparation of coconut husk powders was firstly performed at 105 °C for 8 h. The researchers then continued with the crushing, grinding and sieving processes [87]. The agricultural bio-waste feedstock also needs to be crushed and ground after the drying process. This is to ensure that the fine particle feedstock can encounter more heat energy during the conversion heat treatment process. In an investigation carried out by Biswas et al. [88], corn cob, wheat straw, rice straw and rice husk were dried in the sun, followed by crushing and sieving to obtain the feedstock, with a size between 0.5 and 2 mm. It was demonstrated that drying, crushing, grinding and sieving are the typical processes that are needed, aiming to achieve a high-yield agricultural bio-waste feedstock before converting it into a carbonaceous precursor.

It is clear from the above works that various pre-treatment techniques are used to treat agricultural waste, to control the additional moisture content generated by the humidity of the air and during the cleaning process. Once the raw materials have been pre-treated, they will then be ready for the graphite synthesis process. The method of synthesis is explained in the following section.

## 3. Synthesis Method of Graphite from Agricultural Bio-Waste Material

The ideal method of producing graphite from agricultural biomass can be divided into two main stages [89]. The first is the production of materials with high carbon content from biomass through the carbonization process. This is followed by graphitization, which involves the restructuring of the amorphous carbon structure into crystalline graphitic carbon.

The raw biomass material still has low carbon content. This is due to the phenomenon that is known as “biomass recalcitrance”. The phenomenon describes the natural resistance of plant cell walls to degradation by microbes and enzymes [90]. Lignin and hemicellulose in lignocellulosic biomass are thought to contribute primarily to biomass recalcitrance because they are the polymers that coat the cellulose microfibrils [91]. Therefore, in order to convert biomass waste material into a carbon precursor, pyrolysis is required. After carbonization, the volatiles in the biomass waste are released and the biomass waste is converted into carbon [92]. In addition, to overcome the recalcitrance, the lignocellulosic material is pre-treated with a combination of chemical and structural changes to the lignin and carbohydrates. The pre-treatment process of heating destroys the compact structure of the lignocellulose, which leads to the exposure of the cellulose fibers, which are one of the carbon sources [93]. In the following section, this article will review the crucial role of synthesizing graphite from biomass. The discussion in this session includes the carbonization of agricultural biomass waste materials into carbon precursors and various methods for the graphitization of carbon formed from agricultural biomass waste material.

### 3.1. Carbonization of Agricultural Biomass Waste Material

Proximate analysis can be used to determine the composition of a biomass that burns in a gaseous state (volatile matter), a solid state (fixed carbon) and as an inorganic waste product (ash). As a result, it is critical for the utilization of biomass as an energy source [94]. In addition, through ultimate analysis, one can determine the elemental chemical constituents of the biomass. The most common elemental chemical constituents are carbon, hydrogen, nitrogen and sulfur [95].

Lignocellulosic biomass (LB), which contains cellulose, hemicellulose, lignin and negligible amounts of pectin and ash, has high organic carbon content [96]. In general, most of the hard-shell biomass is LB due to its cellulose content. It is a linear polymer consisting mainly of 3000 to 14,000 glucose monomers linked together by 1,4 glycosidic bonds. This cellulose polymer is normally found in the woody parts of plants and provides the plant with the properties of being hard and insoluble in water [97]. The lignocellulosic decomposition temperature according to thermogravimetric analysis (TGA) is depicted in Figure 2. According to TGA studies, hemicellulose decomposes at a lower temperature (220–315 °C) than cellulose (300–400 °C), whereas lignin decomposes at a higher temperature (150–900 °C) [98,99,100]. Thus, the decomposition of LB is strongly influenced by its composition, as cellulose, hemicellulose and lignin in LB decompose at different temperature ranges.

In the study by Tarelho et al., the carbon concentration of the biochar produced increased with the increasing heating rate. This is due to the fact that increasing numbers of lignocellulosic structures are decomposed and converted into carbon [101]. The same result was reported by Selvarajoo et al., where the carbon content of raw dried citrus peels of 50.85 wt% was increased to 81.80 wt% with the increasing temperature. The increase in the carbon density indicates a greater extent of carbonization and a greater number of volatiles released at higher pyrolysis temperatures. However, when the pyrolysis temperature is increased further, the carbon content decreases slightly. The phenomenon is caused by the pyrolysis temperature exceeding the decomposition temperature of the citrus peels, which excessively decomposes the carbon structure of the citrus peels, causing a drop in carbon concentration [102]. Table 2 summarizes the improvement of the carbon content in the biomass after the pyrolysis process. Studies [101,102,103,104,105,106,107] support the notion that the pyrolysis heat treatment in the charring carbonization process increases the solid carbon content in the biomass material, and, thus, the biomass is converted into a precursor with high carbon content, which is ready for the next graphitization treatment to form graphitic carbon.

However, since the lignocellulosic composition of a biomass has a decisive influence on its decomposition, it has been proposed to analyze the biopolymer content of the biomass before it is subjected to the charring carbonization process. The heating temperature and duration can be set according to the biopolymer composition of the specific biomass material in order to optimize biochar production. Biomass with higher lignin content, for example, requires higher and longer heat treatment because it has a wide range of decomposition temperatures.

### 3.2. Graphitization of Agricultural Carbon

The graphitization process is the process of converting amorphous carbon into graphitic carbon. During the graphitization process, in which the amorphous carbon is heat-treated over a long period of time, the heat energy supports the restructuring of the atomic structure into an ordered crystalline structure, which represents the graphitically structured carbon. The currently most common synthesis method of graphite, which is widely applied in the industry using fossil raw materials, requires a high graphitization temperature of up to 3000 °C [36].

Alternatively, artificial graphite can also be produced from biomass as a carbon precursor [38,108,109,110]. The product obtained from the biomass, e.g., activated carbon and biochar, has high carbon content. A further graphitization process can help to convert the carbon precursor from the biomass into graphite. Graphitic carbon from carbon biomass precursors offers many advantages. Besides being an abundant and inexpensive feedstock, the synthesis of graphitic carbon from biomass requires a lower graphitization temperature, which is beneficial for environmental protection. However, there are various methods that have been used for the graphitization process—for example, graphitization by direct heat treatment, catalyzed graphitization and the addition of activating agents. There is also the Joule heating (Ohmic heating) method and ultrasound-assisted pyrolysis.

#### 3.2.1. Direct Heat Treatment Graphitization

This section is mainly focused on graphitization via the direct heat treatment of biomass waste. Graphitization by direct heat treatment is the most used method because it is the simplest. In some cases, various chemicals, such as catalysts [109,110] and activators [29], are used to increase production or decrease the temperature required for graphitization. However, it is worth mentioning that the heating of biomass has disadvantages as it releases toxic chemicals such as dioxins, dust, carbon monoxide, etc. [108], which further impact the environment. Therefore, alternative methods, such as Joule heating and ultrasound-assisted synthesis methods, which will be discussed in the following section, should be urgently investigated.

##### Effect of Catalyst and Temperature on Graphitization

The catalyst solution is the chemical added to the sample to assist the graphitization of biocarbon by lowering the graphitization temperature. Normally, a metallic solution is widely used in the graphitization field. In the study conducted by Xia et al. [37], monometallic and bimetallic catalysts were investigated to convert biocarbon from bamboo into graphitic carbon. The experiment was conducted in a vertical fixed bed. The effect of the pyrolysis temperature for each of the catalysts used was also investigated. The metallic catalysts used were iron nitrite [Fe(NO_3_)_3_] and either cobalt nitrate [Co(NO_3_)_2_] or nickel nitrite [Ni(NO_3_)_2_]. A BFeCo sample, a BFeNi sample and a monometallic catalyst sample of BFe, BNi and BCo were prepared. All samples were heated in a reactor at temperatures between 550 and 1300 °C for 1 h. The heat-treated samples were then washed with hydrochloric (HCl) acid for 48 h and then rinsed three times with deionized (DI) water and dried [37].

The result showed a high degree of graphitization and the formation of abundant porous graphitic carbon. Because of the presence of Fe-Co alloys and the homogeneous distribution of Fe and Co, the bimetallic Fe-Co catalyst produced more hydrogen (7.51 mmol/g), had a larger pore volume and had a higher degree of graphitization (0.432 graphitization parameter) than the monometallic Fe and Co catalysts. Fe had the highest degree of graphitization and surface area of the monometallic catalysts, while Co had the highest hydrogen yield (7.19 mmol/g biomass). The ideal pyrolysis temperature was 850 °C, which ensured a balance between the biochar porosity and graphitization. Furthermore, the porous graphite prepared with the Fe-Co catalyst demonstrated excellent electrochemical performance for ORR under alkaline conditions, with a half-wave potential of 0.79 V, as well as high stability and excellent electrochemical performance for oxygen reduction [37].

However, this study focuses on a relatively narrow range of catalyst types and pyrolysis temperatures. It would be interesting to see how other catalysts and temperature ranges affect the graphitization process. In addition, the researchers do not address the economic viability of their approach, which would be important for practical applications, given the large-scale synthesis of graphitic carbon in industry. In addition, the issue of the safety of the chemical catalyst for the environment also needs to be considered regarding large-scale applications.

Besides the above, Sun et al. [38] also described the usage of an iron nitrate solution for the catalyzed graphitization of sawdust. This study clearly shows that the temperature is a crucial factor affecting the morphology of Fe and, consequently, the graphitization of the biomass. The pine sawdust was first pre-treated and dried to remove free moisture. The Fe catalyst with molarity of 0 to 7.5 mmol was prepared and mixed with the sawdust. To ensure that the Fe ions were well adsorbed and coated on the surface of the pine sawdust, the mixture was stirred at 80 °C for 2 h and then dried overnight at 100 °C under atmospheric conditions. The sample was then pyrolyzed in a furnace at a temperature between 600 and 800 °C at a rate of 5 °C/min, under a stream of N_2_ at a rate of 50 mL/min [38]. The pyrolysis gas produced at the end of graphitization was directly proportional to the amount of catalyst and the pyrolysis temperature. When the graphitization temperature increased from 600 to 800 °C, it was found that the degree of graphitization increased, while the solid content of the biochar decreased. When the amount of catalyst added was increased threefold, the degree of graphitization of the biochar increased by 17.4%. Increasing the temperature to 800 °C could promote Fe reduction, hasten the dissolution and precipitation of Fe-C and hasten the formation of the graphite structure. By increasing the temperature above 600 °C, the graphitization degree was increased by 76.2% [38].

The above study focused on only one biomass with a precursor. Moreover, the increase in the degree of graphitization after the increase in the catalyst concentration was too low. It is suggested to use a larger range of catalyst concentrations with different temperatures so that the trend of improvement can be seen and predicted. The usage of more than one precursor will also be favorable in order to confirm the effect and feasibility of the catalyst type for different types of biomass waste.

Lignin is one of the biomass waste materials that can be easily obtained. Currently, the industry is pursuing value-added usage in order to strengthen the economy. Via graphitization, as with most biomass waste materials, lignin can also be transformed into graphitic carbon, which brings broad benefits and high-economic-value applications in various fields. According to the investigation by Demir et al. [111], graphitic carbon can be successfully synthesized from lignin using the two-step conversion method, which involves carbonization to form a lignin carbon precursor, followed by the graphitization of the aligning carbon with the help of a catalyst. In the report, carbon was prepared by using the hydrothermal method. To produce biochar, lignin was first carbonized in the presence of water at 300 °C and 1500 psi. The biochar was then graphitized at 900–1100 °C in an inert argon gas at 15 psi using a metal nitrate catalyst. Three catalysts were used, which were iron [Fe(NO_3_)_2_], cobalt [Co(NO_3_)_2_] and manganese nitrates [Mn(NO_3_)_2_]. It was proven that both the heating temperature and type of catalyst influenced the degree of graphitization. A good-quality graphitic carbon was obtained using catalysis by Mn(NO_3_)_2_ at 900 °C and Co(NO_3_)_2_ at 1100 °C. The result was supported by the XRD analysis, as it displayed high crystallinity. Although it was claimed that the graphitic carbon was successfully produced, the micrograph obtained via scanning electron microscopy (SEM) was not clear enough to determine the layering structure of the graphitic carbon. Moreover, the 2D band, which appears after 2500 cm^−1^ in the Raman spectroscopy band, is a significant characteristic of graphitic carbon [110]. Thus, it is encouraged to perform structural scanning of the sample under higher magnification to determine the graphitic layering structure and to perform Raman analysis under a wider wavelength that is able to reveal the 2D band.

A green approach to transforming lignocellulosic materials into highly crystalline bio-graphitic carbon by using the method of the catalyst impregnation of palm waste was evaluated by Jabarullah et al. [25]. The palm kernel shell was immersed and stirred for 48 h in an iron (II) nitrate hexahydrate solution. After drying, the sample was then graphitized at 800 and 1000 °C, respectively, with a heating rate of 5 °C/min, in a nitrogen atmosphere. The acid wash step was performed after graphitization to remove the residual metal catalyst using HCl acid. The same sample preparation steps were repeated while replacing the catalyst with an aqueous solution of nickel (II) nitrate hexahydrate and a hybrid of iron (III) nitrate nonahydrate and nickel (II) nitrate hexahydrate.

From the analysis of XRD and Raman spectroscopy, the sample that was graphitized under 800 and 1000 °C and that was impregnated with iron, nickel or the hybrid iron–nickel catalyst was found to display changes in terms of the microstructure. From the results, the iron-based catalyst was observed to be the most active graphitization catalyst, showing a good degree of graphitization regarding the palm kernel shell. The same was true for the 1000 °C graphitization temperature, which gave the optimum degree of graphitization in this study. A 2θ = 26.5° which is the peak for (002) graphite [112], was recorded for the Fe, 1000 °C graphitized sample. Raman spectroscopy resulted in a lower I_d_/I_g_ value, supporting the notion that the sample formed had a low-defect graphitic structure. However, it was noticed that the control sample, without impregnation by any of the catalysts, showed a disordered microstructure. In the comparison, the authors clearly showed the function of the catalyst in realizing the synthesis of bio-graphitic carbon from palm kernel waste.

The study clearly shows the effect of the catalyst and temperature in the synthesis of bio-graphitic carbon from biomass waste. It is noted that the graphitization process was performed under the flow of inert nitrogen gas. It would be interesting to perform a study in different atmospheres to investigate the effect of the atmosphere on the production of this bio-graphitic carbon.

An increase in the catalyst leads to an increase in the graphitic carbon formed. This can be explained by the fact that when the amount of catalyst is increased, more metal ion dissolution precipitation sites are available for carbon atoms, which promotes the formation and decomposition of the biomass, leading to an improvement in the degree of graphitization of graphitic carbon. As the temperature increases, more thermal energy is provided to convert the amorphous carbon structure into graphitic carbon, as thermal energy is used for the rearrangement of the carbon structure. This further enhances the degree of graphitization. Despite this, various methods can be applied in order to enhance the degree of graphitization. One of the methods that is widely applied is the use of carbon-activating agents, which will be discussed in the following session.

##### Effect of Activated Agent

Thermochemical activation is of great interest because of its wide application. In some industries where a porous structure is favorable, an activating agent is a promising option, as it helps to improve the mesoporous structure of the graphitic carbon.

Wang et al. [39] reported on the chemical activation graphitization of pomelo peel with potassium hydroxide (KOH) and nickel acetate (C_4_H_6_NiO_4_). First, the pomelo peel was cut and washed with deionized water. The dried pomelo peel was then allowed to react with concentrated sulfuric acid for 4 h at 120 °C to dehydrate the pomelo peel. The dehydrated pomelo peel (DPP) was then mixed with the activator, KOH, and nickel acetate in a mass ratio of 1:1:0.5. Under an inert argon atmosphere, the mixture was subjected to pyrolysis heat treatment at 800 °C for 4 h. The process ended with an acid wash to remove the ions and the remaining impurities of the graphitic carbon [39]. The results indicated that an excellent and tunable, hierarchical porous structure was found on the DPP using the one-step activation graphitization procedure. The DPP carbon product not only had a graphitic structure but also has a unique microscopic pore structure. Due to the above structure, the DPP graphitic carbon was able to achieve a high specific capacity and excellent rate capacity in the three-electron system. The authors also found that symmetrical DPP graphitic carbon supercapacitors with aqueous electrolytes or ionic liquid electrolytes had better energy densities than those described in the literature [39]. The success of this study further confirms that biomass-derived graphitic carbon might offer a great contribution to the supercapacitor field in the near future. However, this study focuses on a single type of biomass (pomelo peel), so it is unclear whether the results are generalizable to other types of biomass. In addition, the researchers did not compare their method to other methods of adjusting the hierarchical porous structure and degree of graphitization of biomass-derived carbon, so it is unclear whether their approach is competitive.

Tan et al. [40] used K_2_FeO_4_ as the activator in their study to produce a three-dimensional, highly graphitic porous biomass carbon (HGPBC) using a dandelion flower stem. The same activator was used by Gong et al. [41], where high capacitance (222.0 F g^−1^ at 0.5 A g^−1^) was produced. As with all biomass material pre-treatments, the dandelion flower stem was rinsed and dried to remove impurities and then immersed in the K_2_FeO_4_ activator solution. The mixture was stirred at 70 °C for 12 h until the solvent evaporated. The impregnated dandelion flower stem was then heat-treated at 700 °C for 2 h. As in the previous report, the sample was washed with HCl acid, and then washed with DI water and dried overnight. The HGPBC formed in the study showed a low degree of defects and disorder. This was confirmed by the I_d_/I_g_ result from Raman spectroscopy, as the HGPBC had a lower I_d_/I_g_ value of 0.906, while this value was 0.906 for pristine carbon, which has no activation. With a specific surface area of 780.4 m^2^ g^−1^, the obtained HGPBC achieved high specific capacitance of 309 F g^−1^ at a current density of 0.5 A g^−1^ and a high energy density of 14.22 Wh kg^−1^ at a power density of 218.8 W kg^−1^ [40].

In the study by Tan et al. [40], the focus was exclusively on the use of dandelion flower stems as a biomass source. The dandelion flower stem is found relatively less frequently compared to other agricultural wastes, such as coconut shells, palm kernel shells, hard nut shells, etc. Moreover, the carbon content of the carbon precursor, which significantly influences the quality of the graphitic carbon produced, must be determined. In Tan et al.’s [40] study, the ultimate analysis to determine the carbon content of the raw dandelion flower stem was not performed. Thus, the original weight percentage of carbon content is not known. Therefore, it is unclear whether the results are generalizable to other common biomass materials. In addition, the researchers did not compare their method to other approaches to producing porous carbon materials with high graphite content, so it is unclear how competitive their approach is.

Ning et al. [113] used macadamia shells to produce graphitic carbon, which was then used as a cathode oxygen reduction catalyst in microbial fuel cells. A combination of K_2_FeO_4_ and urea was used as an activator in their research. In the process, the macadamia shell was placed in a tube furnace and carbonized at 300 °C under a nitrogen flow for approximately 30 min. First, the carbonized macadamia shell biomass was ground and sieved on a 200-mesh screen. It was immersed in 500 mL of a 10 g L^−1^ urea solution for 8 h to mix completely, and then dried at 60 °C for 48 h after being stirred continuously for 8 h. It was then immersed in a solution of 10 g/200 mL K_2_FeO_4_ for 8 h, before the dried product was magnetically stirred for another 8 h. The dried solid mixture was then placed in a tube furnace and heated to the set temperature under nitrogen conditions, with a heating rate of 5 °C min^−1^ for 2 h.

The catalyst prepared at 750 °C exhibited a large specific surface area (1670.3 m^2^ g^−1^) and a graphite structure. The catalyst exhibited excellent oxygen reduction reaction (ORR) with an onset potential of 0.172 V and half-wave potential of −0.028 V (vs. Ag/AgCl) in a neutral medium, and the ORR electron transfer number was 3.89 [113].

However, no 2D peak was found for the sample in Raman spectroscopy. The 2D peak is the characteristic feature of graphitic carbon in Raman spectroscopy analysis [110]. Therefore, further tests should be performed to detect the presence of graphite in the product.

To prepare nitrogen-doped, biomass-derived, hierarchically porous carbon materials (HPCs), Cao et al. [114] proposed a facile one-step method. Nitrogen-containing compounds (such as NH_4_Cl, (NH_4_)_2_CO_3_ and urea) were used in the study, acting as the activator and dopant. They started by mixing a nitrogen additive solution with camellia pollen, which was then pyrolyzed at a heating rate of 5 °C/min to 800 °C and maintained for 2 h. The pyrolyzed product was then acid-washed using 20 wt% HCl acid, DI water and C_2_H_5_OH to remove the alkaline products.

From the study [114], the NH_4_Cl was found to be a porogen and showed the least collapsed pollen grains. The study also suggests urea to be the most effective N dopant. From XRD, all HPC samples were found to have a broad peak at 24–26°, assigned to the graphitic (002) plane [112], confirming the coexistence of ordered graphite layers and turbostratic graphite. Samples prepared by urea and NH_4_Cl exhibited an additional slight weak peak at 42.6–43.8°, which is the (100) plane of graphite [115]. All samples showed the defect-induced D (1330 cm^−1^) and the crystalline graphite G (1585 cm^−1^) bands in Raman spectroscopy, suggesting the formation of high-defect graphite-like materials [116,117]. The good conductivity, resulting in HPCs that are activated with N-containing compounds, is in line with the high I_G_/I_D_ intensity ratio, which means a higher degree of graphitization as compared to inactivated HPCs.

It was mentioned that the presence of graphitic carbon in the product was observed. However, the peak shown in XRD, described as the (002) plane and (100) plane of graphite, is not convincing as the peak formed is too broad for the (002) plane but too weak for the (100) plane. A broad peak in XRD indicates an amorphous structure, while graphitic carbon should show a sharp and high-intensity peak. Thus, Cao et al. should provide more confirmatory tests to prove their success in synthesizing graphitic carbon in their sample.

Yan et al. [118] developed a novel “wet” activation approach to produce porous carbon from pistachio shells by using synergistic supercritical water and KOH. The benefits of this method are that it uses less activator and is more facile and greener, which lowers the time consumption, with no harmful by-products formed.

The preparation starts by placing the pistachio powder, different ratios of KOH and 50 mL of DI water into the reaction tube of the supercritical reactor kettle. The reactor kettle was heated at the rate of 10 °C min^−1^ to 400 °C and for approximately 20 min. The resulting product was then rinsed and dried, followed by heating in an alumina crucible at 900 °C for 2 h at a heating rate of 10 °C min^−1^, and then acid-washed to remove impurities. Overall, the sample produced at a sample mass ratio to KOH of 5:1 showed the highest carbon content among the treated biochar samples. Broad XRD peaks could be observed at 26° (002) [112] and 44° (100) [115] with low intensity, revealing the amorphous structure of the samples with a degree of graphitization. This was confirmed by the Raman spectrum, which presented two peaks at around 1320 and 1580 cm^−1^, corresponding to the D-band and G-band, respectively. The sample also had a high I_g_/I_d_ value, which is the reason that the sample possesses high electrical conductivity due to its high graphitization structure. The electrode prepared using the optimum sample showed outstanding capacitance of 232 F g^−1^ (260 F cm^−3^) at 0.5 A g^−1^, exerting high capacitance retention of 144 F g^−1^ (161 F cm^−3^) at 100 A g^−1^ [118].

The result of this study showed that the weak graphitic structured carbon produced from the pistachio shell using a KOH activator displays great potential in the electric conductivity field. It also shows that KOH is an effective activator in producing graphitic carbon. Improvements should be made regarding the concentration of the KOH activator in a coming study so that more graphitic carbon might be formed, which may allow even better electrochemical performance for supercapacitors.

Overall, the use of activators in graphitization shows great success in producing mesoporous graphitic carbon to contribute to energy storage applications in the future. The proposed feedstocks are biomass wastes that have relatively low carbon content. Therefore, it is proposed to apply the same processing method to biomass with higher carbon content to achieve even higher quality in the graphitic carbon produced.

#### 3.2.2. Joule Heating Method

In the study conducted by Gelfond et al. [119], carbon fibers were obtained from bamboo. After a stabilization and carbonization process, the carbon was extracted from the biomass. The carbon fibers were successively Joule-heated at ∼2000 °C for 20–30 s to form graphitized carbon fibers. The process was carried out in an argon box by holding the carbon fibers between two copper plates and fixing them at both ends with silver lacquers. The copper plates were 1.5 cm apart, and power cables were attached to each plate. The voltage was gradually increased until the fibers maintained their temperature (glowed with constant brightness), and then it was immediately increased to the target voltage, which was between 40 and 95 V. The target voltage was held for 5–120 s or until the fiber cracked [119]. From this study, the result showed that the produced graphitized carbon fibers had nanofibril alignment and high electrical conductivity of 25,300 ± 6270 S/m.

In Jiang et al.’s [120] study, graphene oxide (GO) was used to improve the electrical conductivity by forming a film with a lignin biomass. First, alkaline lignin with low sulfonate content was mixed with GO in a 1:1 ratio. After adding DI water, the solution was allowed to react for 2 h under ultrasonic action. The solution was then poured into a Petri dish and dried at 60 °C to obtain the GO–lignin film. In this study, the lignin content was set at 50% to maximize the possible lignin content. The GO–lignin film was then carbonized at 873 K for 2 h under an argon-protective atmosphere using the Joule heating process.

The lignin-based graphitic carbon exhibited electrical conductivity of 4500 S/cm, which is ultrahigh. The composite film that heated ~2500 K within 1 h had an improvement of over 700 times in terms of electrical conductivity as compared to a film that was carbonized at 873 K. Through HRTEM scanning, it could be observed that the short-range ordered carbon and extended graphitic structure could be found in the film. This was supported by the XRD and XPS spectra [120].

In this report, we can conclude that the Joule heating method can successfully produce high-quality graphitic carbon, which can also be applied as an electrical conductor. A great benefit of the Joule heating method is the production of minimal defects in lignin-based carbon, which cannot be achieved when using the normal graphitization method. Thus, various types of biomass material should be used in exploring the Joule heating method to extend its benefits to different, low-cost biomass types, to achieve the cost-effective synthesis of graphitic carbon.

From the mentioned papers [119,120,121,122,123], the Joule heating method has successfully produced graphitic carbon with a low cost but high electrical conductivity. Due to their cost-effectiveness, they are a promising component in smart construction or conductive thermoplastics and resins. According to the literature, unlike natural bamboo fibers, which degrade over time, carbon fibers can permanently bind 0.85 t of carbon per hectare of bamboo per year [119]. This is significantly more carbon than is sequestered in carbon fibers made from wood cellulose. In addition to carbon fibers, the lignin produced during delignification is a valuable by-product for the energy and health industries.

The flash Joule heating (FJH) method has been attracting the attention of researchers due to its benefits in performing high-quality graphitization on carbon products in a short time range. Mohamed et al. [121] performed a study comparing the FJH method with the traditional high-temperature heating (HTH) method, which is widely utilized by the industry. The carbon precursor applied in this study was graphitized coal, which was prepared in two main steps, which were carbonization and graphitization. In the first stage, the raw coal powder underwent a carbonization process at 600 °C to convert it into activated carbon with high carbon content. The process’ duration was 6 h, with a 5 °C min^−1^ heating rate under an argon gas flow. Each gram of coal-activated carbon was then mixed with 100 mL of ethanol and 5 mmol nickel chloride (NiCl_2_). The mixture was stirred for 1 h at 600 rpm, followed by the drying process.

In the second step of the study [121], the impregnated coal carbon was graphitized using the HTH method at 1400 °C with a 5 °C min^−1^ heating rate for 10 h under an argon gas flow, followed by an acid wash. For the FJH method, a weight ratio of 10:90 of conducting additive carbon black was mixed by grinding using a mortar and pestle with the coal powder. A similar heating method was performed with the capacitor bank (capacitance of 60 mF) used to provide a direct discharge current with maximum voltages up to 400 V. First, 300 V was applied to the sample for 5 s; then, the sample was ground and acid-washed to remove impurities. The high-resolution translation electron microscopy (HR-TEM) showed nickel-impregnated coal carbon that was produced using the HTH method (HTC-Ni) consisting of graphitic carbon and amorphous carbon that were mixed together. After catalytic graphitization using FJH, the nickel-impregnated coal (FHC-Ni) displayed a graphitic structure under transmission electron microscopy (TEM) and HR-TEM, which was separated by an interlayer of 0.338 nm, which aligned with the result from XRD. Therefore, it can be concluded that both the HTH and FJH methods could successfully synthesize graphitic carbon with the help of a nickel activator. However, coal is a precursor that has very rich carbon content. An alternative biomass with different carbon content should be used to test the feasibility of this method as the success rate will also depend strongly on the carbon content of the raw materials.

The pyrolysis process using a conventional electric furnace brings with it the problem that the heating or cooling process is too slow, and the heat is not sufficiently utilized. To overcome this, research is now focusing on a new, fast and effective heating method involving pyrolysis by Joule heating [122]. A carbon fiber cloth (CFC) was soaked in acetone for approximately 15 min and then washed with DI water. For the electropolymerization of PANI, a mixture of sulfuric acid (H_2_SO_4_) and pure aniline was used as the electrolyte, and the cyclic voltammetry technique (CV) was employed. A graphite rod and an Ag/AgCl electrode were used as counter- and reference electrodes, respectively. The prepared PANI @CFC was washed and dried, and then the pyrolysis apparatus, consisting of a piece of PANI @CFC precursor (1.0 × 2.0 cm^2^), two pieces of fused silica, four nickel plates, two power cables with alligator clips and a DC power supply, was placed in a stainless-steel glove box (DECO-VGB-304-2-O, Changsha Deco Equipment Co., Ltd., China) and fitted with a thermal baffle. The DC power supply was connected to another end. Inside the sealed glove box, which was equipped with a vacuum pump, an inert nitrogen environment (~1.0 atm) could be created during pyrolysis by Joule heating. The pyrolysis voltage and duration were set to 1.0–6.0 V and 5.0–15.0 min, respectively.

From the result [122], three Raman peaks at ~1350 cm^−1^ (“D”), ~1580 cm^−1^ (“G”) and ~2680 cm^−1^ (a weak “2D”) were seen, which reflect the characteristic Raman reactions of carbon material and demonstrate that the polyaniline was indeed converted into a carbon-based material after pyrolysis by Joule heating. The “D” peak and the “G” peak separately represent the defect peak and the graphite peak, and their integral intensity ratio (IG/ID) is generally indicative of the degree of graphitization [124]. The lattice fringes in the high-resolution TEM of the granules, which had spacing of ~0.35 nm and corresponded to the (002) lattice plane of the graphite carbon, proved that the pyrolysis products contained graphite carbon. From the result, the pyrolysis treatment with Joule heating had little effect on the graphitization degree of CFC, so the increased graphitization degree of the sample could be attributed to the pyrolysis products coated on the CFC.

In the study by Dong et al. [123], graphite was regenerated in lithium-ion batteries at the end of their life with excellent electrochemical properties via the fast, efficient and environmentally friendly flash Joule heating (FJH) method. The spent graphite powder was removed from the copper foil by placing the anode plate removed from the waste battery in 50 mL of deionized water (DI) and sonicating it for 1 min; it was then dried at 80 °C for 12 h; then, 100 mg of spent graphite was added to the reaction vessel, after which the graphite electrodes were inserted at both ends. After charging several parallel capacitors (2, 5 and 10) to 200 V, the energy was immediately released on the sample. The phenomenon of “flashing” could be observed. The final product (F-RG) was collected as an anode for new LIBs. For the regeneration of spent LiFePO4, the cathode was treated and disassembled by the same method. The spent graphite separated from the smartphone battery was treated by the same method.

Under constant pressure and in an air atmosphere, graphite was rapidly regenerated within 0.1 s, with no pollutant emissions. FJH provides a high current for repairing defects and reconstructing crystal structures in graphite. Regenerated graphite has excellent multiplier and cycling performance (350 mAh g^−1^ at 1 °C with 99% capacity retention after 500 cycles). The cost of recycled graphite, which has the same performance as new graphite, is only 77 CNY per ton [123].

In both of the studies [101,103], the carbon precursor material used was a commercial one. It is worth investigating the replacement of the carbon precursor using biomass waste so that the processing cost for this fast and high-quality Joule heating pyrolysis method can be further lowered using biomass waste.

Although the temperature is relatively high in this newly introduced method, the short heating time should nevertheless not be underestimated, as it will contribute significantly to the mass production of graphitic carbon in the future. This was conducted in a laboratory setting, and it remains to be seen whether these carbon fibers can be produced on a large scale at an affordable cost. In addition, the study did not examine the long-term stability of the fibers or the performance in real-world carbon sequestration applications. Further research is needed to explore their feasibility and effectiveness at scale to exploit their advantages in rapid production while improving it in terms of the heating temperature.

#### 3.2.3. Ultrasonic-Assisted Graphitic Carbon Synthesis Method

Ultrasound technology is one of the most well-known technologies in the research field in at present, as it has proven to be very useful in various fields. Xu et al. [125] present a novel method to produce graphitic carbon from biomass waste, e.g., wheat straw. Graphitization was mainly divided into three steps, namely the degradation of lignin, graphene formation and the graphitization process. A frequency of 16 to 20 kHz was applied to the cut wheat straw under atmospheric pressure. It was then ultrasonically reacted for 30 min and three repetitions [125].

From the report, as confirmed by X-ray diffraction, a clear diffraction peak was found around 2θ = 26.6°. Raman spectroscopy revealed an I_d_/I_g_ value of 1.06, proving the presence of graphite in the ultrasonicated wheat straw pulp. However, a limitation of the research is that the content of graphitic carbon formed was only approximately 4.5%, which is very low [125]. Moreover, the method used to derive the graphitic carbon from the mixture was not mentioned in the study.

Teng et al. [126] also applied the ultrasound-assisted method to fabricate high-performance carbon-based supercapacitors. In this study, garlic shells were used as carbon precursors. Common pre-treatments were applied, including washing and drying of the biomass, namely garlic peels, then placing them into a pulverizer to be ground for 3 min. A siever was used to obtain the 178 μm fine garlic peel powder (GPP). It was then pyrolyzed in a tube furnace at 600 °C for 2 h. KOH was mixed with the GPP at a ratio of 4:1 and impregnated with an ultrasonic disperser at 40 kHz and 500 W for a duration of 0–9 min. The sum of the ultrasonic-assisted and static impregnation time was kept constant at 30 min, and the entire impregnation process was carried out at 65 °C. The mixture was heat-treated in an activated atmosphere furnace at 800 °C for 2 h at a heating rate of 5 °C min^−1^ and an inert gas flow rate of 0.04 m^3^ h^−1^ each. The sample was then washed with 1 M HCl, and the dried sample was obtained as garlic-peel-based porous carbon (GBPC).

As reported by Teng et al. [126], the structure and electrochemical properties of 3D layers of porous carbon are significantly improved after ultrasonic-assisted impregnation over a period of time, being the best at 6 min. Cavitation loosens the surface adhesion of the carbonized product, leading to the homogeneous dispersion of potassium vapor in the garlic peel, resulting in more micropores (0.5–0.8 nm). The specific surface area (SSA) of GBPC increases from 2548 to 3887 m^2^ g^−1^ and the specific capacitance increases from 304 to 426 F g^−1^ at a current density of 1 A g^−1^ in a two-electrode test system. At the same time, the energy density and cycle performance are also improved, which is due to the rational structure. However, based on the results of X-ray diffraction analysis and Raman spectroscopy, no significant graphitic carbon is found in the GBPC. Although the method successfully improved the capacitance of the porous carbon, a structured graphitic carbon will still be more favorable in terms of specific capacitance.

There are limited studies on the ultrasonic-assisted chemical synthesis of graphitic carbon. Therefore, for the following discussion, similar studies to produce carbon-based materials using the ultrasonic-assisted method will be discussed. Bora et al. [127] evaluated the synthesis of activated carbon (AC) by using low-quality sub-bituminous coals. The raw coals were first cleaned and dried. Then, for the oxidation of the coal sample, hydrogen peroxide (20–30% H_2_O_2_) was added drop-wise from the burette. At the same time, the coal slurry continued to be stirred in ice-cold conditions to control the exothermic reaction. The reaction mixture was ultrasonicated (20–40 kHz) at room temperature for 5–6 h and then cooled in an ice-water bath. They then neutralized the mixture by adding ammonia drop-wise until pH 7 was reached. It was then washed with distilled water, and the residue was dried in the oven. The dried oxidized coals were heated at 300 °C in a muffle furnace for 1–2 h in the absence of air to soften and deform the coals. KOH and KOH + NaOH in a mass ratio of 2:1 were used as activating agents for the plastic charcoal. The activating agents (KOH and KOH + NaOH) were mixed with the plastic charcoal in the solution phase. The reaction mixture was placed under ultrasonic irradiation (20–40 kHz) for 1–2 h. To remove water-soluble, basic and alkaline components, each of the reaction mixtures was washed with hot deionized water to the neutralization point. The remaining portion of the solution was oven-dried at 110 °C for 2 h. Finally, the ultrasonicated char samples were charred at 800 °C in a muffle furnace for approximately 2 h. Then, the synthesized product was washed with 1 M HCl solution followed by distilled water until neutralization to remove the remaining inorganic impurities.

The result shows [127] that the adsorption properties, energy density and power density of the ACs were improved by the application of low-energy (40 kHz) ultrasonic irradiation via the formation of a high surface area and surface porosity. Through XRD, it was shown that the ACs had a partially graphitic structure, which matched with the JCPDS card number 75–2078. The peak located at 2θ  =  20–25 corresponded to the (002) plane of carbon-containing aromatic layers with a stacked structure. With the presence of graphitic structures, the electrochemical performance of the supercapacitor cell was improved. Since the graphitic carbon structure was found in the sample product, it is suggested to alter various parameters in order to produce more content of graphitic carbon, which will greatly improve the electrochemical properties.

A greater focus can be placed on the ultrasound-assisted graphitic carbon synthesis method, as Xu et al.’s [125] research has shown that graphite is successfully produced. Regarding the research of Teng et al. [126], no graphitic-phase carbon was found. There is still a large gap regarding the process and the synthesis parameters in the field of ultrasonic-assisted graphitization processes. Therefore, research on ultrasonic-assisted graphitic carbon synthesis methods is insufficient as it is still a very new method. Table 3 shows the summarizes of synthesis methods of graphitic carbon from bio-waste materials and its results.

## 4. Summary and Future Works

From this review article, there are a few weaknesses in the research and we should focus more on the synthesis of biomass-based graphitic carbon to develop the field further.

It is more important to determine the carbon content of the biomass before starting, to avoid wasting resources.The workability of the study in terms of large-scale production is not known, as research is mostly performed on a laboratory scale.The safety of catalysts used in regard to large-scale production should be considered. Will this cause other environmental issues, e.g., creating more harmful waste or improper disposal, causing soil and water pollution?There is insufficient study of advanced graphitic carbon synthesis as research mostly focuses on the common use of one-step heating graphitization.

To realize the synthesis of graphite from biomass waste with lower costs and greater environmental friendliness, several future works could be pursued.

Identification and utilization of novel biomass waste materials that are abundant, have high carbon content and can be easily processed.Development of cost-effective and environmentally friendly techniques for the purification of carbon precursor materials.Optimization of the carbonization and graphitization processes to increase efficiency and reduce energy consumption.Investigation of the properties and performance of the resulting graphite materials for different industrial applications.Development of new catalysts or additives that can improve the quality and yield of graphite.Recycling and reuse of waste products generated during the synthesis process to minimize waste and reduce the overall environmental impact.Study the feasibility of realizing the production of biomass-based graphitic carbon in the industry for large-scale production.Implementation of life cycle assessment (LCA) studies to evaluate the environmental sustainability of the synthesis process and identify areas for improvement.

Overall, these future works would help to advance the synthesis of graphite from biomass waste, making it a more viable and sustainable alternative to traditional petroleum-based methods. From the review, it can be concluded that the carbonization process, followed by the graphitization method, offers a great contribution to further increasing the carbon content of agricultural bio-waste materials. However, in order to improve the synthesis of graphitic carbon without harming the environment, an environmentally friendly catalyst that can be used during the graphitization process needs further investigation.

## Figures and Tables

**Figure 1 materials-16-03601-f001:**
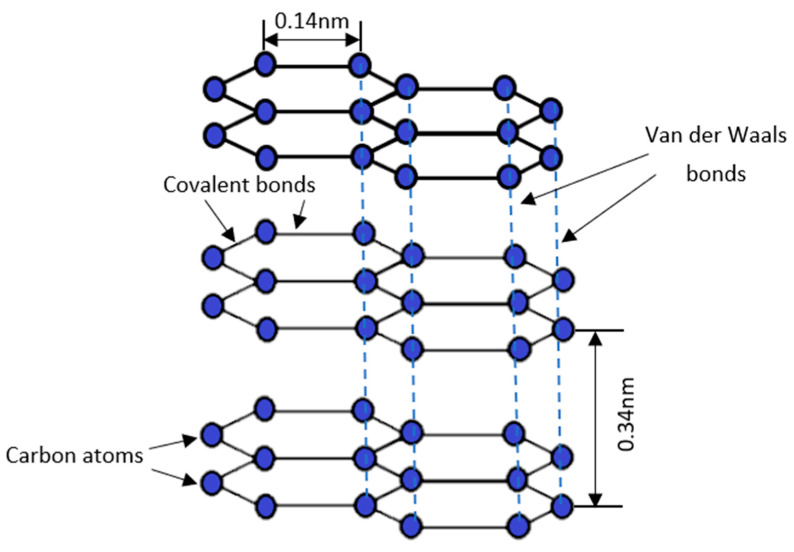
Schematic illustration of graphite structure.

**Figure 2 materials-16-03601-f002:**
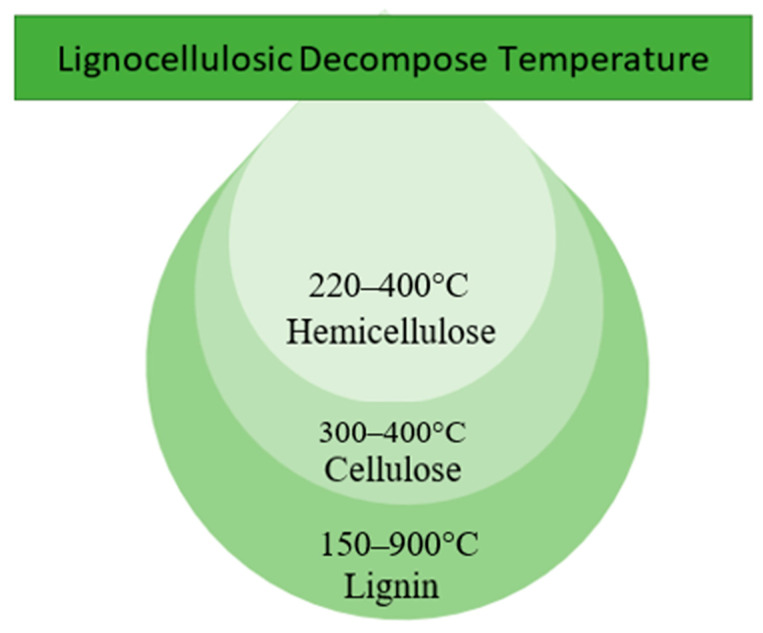
Lignocellulosic decomposition temperature according to TGA analysis.

**Table 1 materials-16-03601-t001:** Proximate and ultimate analysis of various agricultural bio-waste carbonaceous precursors.

Agricultural Waste	Proximate Analysis (% *w*/*w*)			Ultimate Analysis (% *w*/*w*)					Reference
	Moisture Content	Ash Content	Volatile Content	C	H	N	O	S	
Coconut shell	12.55	0.42	NA	51.60	5.60	0.10	42.70	0.00	[78]
Macadamia nut shell	10.00	0.40	71.0	57.50	5.95	0.33	36.20	0.06	[79]
Coconut coir	9.50	9.30	69.80	42.00	4.85	0.42	40.50	0.13	[80]
Kenaf	8.35	16.32	64.20	39.20	5.12	0.35	45.60	0.22	[80]
Rice husk	9.40	13.20	62.00	37.80	4.73	0.45	43.50	0.17	[80]
Oil palm frond	6.00	1.00	76.00	42.88	7.06	0.52	49.54	NA	[81]
Palm kernel shell	8.44	8.79	82.58	50.42	9.74	0.52	39.32	NA	[81]
Chinese chestnut shell	NA	0.00	73.23	46.68	5.94	0.62	46.76	NA	[82]
Bamboo	NA	1.02	82.08	54.30	5.57	0.21	38.91	NA	[82]
Jatropha shell	NA	6.69	70.94	47.35	5.88	1.00	39.08	NA	[82]
Cotton stalk	NA	3.14	80.54	47.12	6.25	0.57	42.79	NA	[82]
Saw dust	NA	0.42	85.14	50.64	6.00	0.07	43.29	NA	[83]
Raw straw	NA	9.98	74.39	50.01	5.54	0.81	43.64	NA	[83]
Apricot kernel shell	9.71	0.94	73.84	46.88	6.38	0.25	45.45	0.00	[84]
Macadamia nut shell	NA	2.51	77.68	49.15	5.51	0.59	42.12	0.12	[85]

**Table 2 materials-16-03601-t002:** Results of improvement of carbon content after pyrolysis process.

Biomass Material	Carbonization Temperature (°C)	Raw Carbon Content (wt%)	Highest Carbon Content after Pyrolysis (wt%)	Reference
Pulp and paper biological sludge	525	NA	47.00	[101]
Citrus peels	500	50.85	81.80	[102]
Douglas fir sawdust	500	47.70	82.76	[103]
Wheat straw	500	43.23	67.66	[104]
Spent coffee ground	500	46.97	76.78	[104]
Brewery grains	500	48.80	70.86	[104]
Rice husk	600	42.50	58.60	[105]
Bamboo	700	48.13	86.34	[106]
Palm kernel shell	850	50.73	77.43	[107]
Wheat straw	850	43.00	55.83	[107]
Pine sawdust	800	46.73	91.73	[107]

**Table 3 materials-16-03601-t003:** Summary of different synthesis methods of graphitic carbon from biomass waste material.

Biomass Material	Synthesis Method	Media	Temperature	Results	Ref.
Bamboo	Direct heating (catalyst and temperature)	Fe(NO_3_)_3_ Co(NO_3_)_2_ Ni(NO_3_)_2_	550–1300 °C 1 h	- Large specific surface area - Abundant micropores - High graphitization degree - Excellent electrochemical performance	[37]
Pine sawdust	Direct heating (catalyst and temperature)	Fe(NO_3_)_3_	600–800 °C Nitrogen atmosphere	- The graphitization degree increased by 76.2% when the catalyst increased - Increase in the formation of graphite structure with temperature	[38]
Coal	Direct heating (catalyst and temperature)	NiCl_2_	1400 °C 10 h Argon atmosphere	- A mixture of amorphous and graphitic carbon found under HR-TEM	[121]
Palm kernel shell	Direct heating (catalyst and temperature)	Fe(NO_3_)_3_ ·9H_2_O H_12_N_2_NiO_12_ Fe(NO_3_) ·6H_2_O	800 °C, 1000 °C Nitrogen atmosphere	- Low-defect graphitic structure	[25]
Lignin	Direct heating (catalyst and temperature)	Fe(NO_3_)_3_ Co(NO_3_)_2_ Mn(NO_3_)_2_	900–1100 °C 3 h Argon gas at 15 psi	- Good-quality graphitic carbon was obtained using catalysis by Mn(NO_3_)_2_ at 900 °C and Co(NO_3_)_2_ at 1100 °C	[111]
Pomelo peel	Direct heating (activated agent)	KOH C_4_H_6_NiO_4_	800 °C 4 h Argon atmosphere	- Formed hierarchical porous structure - High specific capacity - Better energy density as a supercapacitor	[39]
Dandelion flower stem	Direct heating (activated agent)	K_2_FeO_4_	700–900 °C 2 h Nitrogen atmosphere	- Graphite formed with a low degree of defects and disorder - High specific capacitance - High energy density	[40]
Bamboo	Direct heating (activated agent)	K_2_FeO_4_	800 °C 2 h Argon atmosphere	- Large specific surface area - Abundant micropores - High graphitization degree	[41]
Camellia pollen	Direct heating (activated agent)	NH_4_Cl, (NH_4_)_2_CO_3_ urea	800 °C 2 h	- Hierarchical porous surface carbon structure formed - High-specific-capacitance electrode from biomass graphitic carbon	[114]
Pistachio nut shell	Direct heating (activated agent)	KOH	900 °C 2 h	- Weak graphitic carbon formed - High capacitance of 232 F g^−1^ (260 F cm^−3^) at 0.5 A g^−1^	[118]
Bamboo	Joule heating	-	~2000 °C 20–30 s Argon atmosphere	- Nanofibril alignment - High electrical conductivity (25,300 ± 6270 S/m)	[119]
GO/lignin film	Joule heating	-	~2500 K 1 h Argon atmosphere	- Ultrahigh electrical conductivity (4500 S/cm) - Improvement of over 700 times in terms of electrical conductivity - Short-range ordered carbon and extended graphitic structure	[120]
Coal	Joule heating	NiCl_2_	300 V 5 s Vacuum atmosphere	- Graphitic structure found in TEM, HR-TEM - An interlayer distance of 0.338 nm observed between the sample layers	[121]
Carbon Fiber Cloth (CFC)	Joule heating	-	1.0–6.0 V 5.0–15.0 min	- Pyrolysis products contained graphite carbon - Under TEM, granules had spacing of 0.35 nm	[122]
Used lithium battery graphite	Joule heating	-	200 V	- Low-cost recycled new graphite - Regenerated graphite had excellent multiplier and cycling performance	[123]
Wheat straw	Ultrasonic assisted	Ammonia (5%) Sodium dodecysuphate (0.5%) Silicate (3.5%) Magnesium sulfate (1.5%)	16 kHz to 20 kHz 30 min, thrice	- A clear diffraction peak was found around 2θ = 26.6° - I_d_/I_g_ value of 1.06, proving the presence of graphite - Graphitic carbon formed was too low (only approximately 4.5%)	[125]
Garlic shell	Ultrasonic assisted	KOH	Impregnated at 40 kHz 500 W Heated at 800 °C, 2 h with inert gas flow	- Improved energy density - Improved specific capacitance - No significant graphite peak in Raman spectroscopy (missing 2D peak)	[126]
Low- quality sub-bituminous coals	Ultrasonic assisted	KOH and NaOH	20–40 kHz 5–6 h Heated at 800 °C, 2 h	- Little graphitic structure found in the activated carbon - Improved the electrochemical properties (capacitance)	[127]

## Data Availability

Not applicable.

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
