# Peer review of "Recent Advances in Synthesis of Graphite from Agricultural Bio-Waste Material: A Review"

_materials, 2023, doi:10.3390/ma16093601_

Round 1

Reviewer 1 Report

This manuscript is topic of review on “Recent advance in synthesis of graphite from agricultural biowaste material: A review”.  I would like to say that the review is pretty impressive, unourtunately unacceptable without majör revision. comments are attached as a file

Author Response

Thank you for your comments. Please find my respond at the attachment.

Reviewer 2 Report

In this review article, the authors collected the previous research that had been done in synthesizing crystalline graphite, the biomass pretreatment methods to optimize the conversion of biomass waste material into carbon precursor, the gap discovered in the previous literature was also suggested.

It is very original and interesting. It can be published like it is.

Author Response

Thank you for your review. No response to Reviewer 2 has been made since it is accepted as it is. Thanks again

Reviewer 3 Report

Review of the Manuscript No Materials-2347481-peer-review-v1 (1)

This work is dedicated to the recent advance in synthesis of graphite from agricultural waste. Different production methods are examined. The effect of different parameters - temperature, pressure, activating agent, etc. - on the synthesis procedure are discussed. The review is profound and extensive, with more than 100 actual references. The work has significant contribution towards environmental protection.

I recommend acceptance of the paper after minor revision.

Meanwhile I have some notes, questions and recommendations.

1. English text should be improved. Introduction should be rewritten.

Introduction, Page 2, Row 59-67

“As mentioned, carbon has four valence electrons. Due to its valency, it is able to form various allotropes. An allotrope is a form when an element exists in more than one type of crystalline form. Allotropes have identical chemical properties as it is still formed by the same element. However, it varies in terms of physical properties as the allotropes exist in different structures [6]. For example, diamond is well known to be a hard and inert material [7]. This is due to the reason that it forms a strong covalent bond between the carbon atom, which then turns out to be a 3D arrangement to its structure [8]. While for graphite has a soft and slippery property as the bonding force between the layers in graphite is easily released [9].”

Should be substituted with

“As mentioned, carbon atom in excited state has four valence electrons, and it is able to form various allotropes. There is allotropy when an element exists in more than one type of crystalline form. Allotropes have identical chemical properties, however they have different physical properties, as the atoms of the element are bonded together in various structures [6]. For example, diamond is well known as very hard and inert material [7]. This is due to the reason that there is a strong covalent bond between the carbon atoms, bonded together to form a cubic lattice of 3D tetrahedra [8]. In the same time, the carbon atoms in graphite are bonded together in sheets of hexagonal lattice - bonding between layers is relatively weak van der Waals bonds, which allows the layers to be easily separated and to glide past each other.”

2.  Introduction, page 2, Row 57-58

“For example, the allotropes of carbon include diamond, graphite, lonsdaleite, and fullerene [3, 4, 5].”

Should be replaced by

“For example, the allotropes  of carbon include diamond, graphite, amorphous carbon, lonsdaleite, carbon nanotubes and fullerene [3, 4, 5].”

3. In Summary the conclusions are not very well defined. There should be some more clear conclusions. For example, there is no conclusion which methods of synthesis are better and why. Summary section should be rewritten.

Author Response

Thank you for your comments and suggestions. Please find my response at the attached file. Thanks again

Round 2

Reviewer 1 Report

I am pleased to inform accept of revised manuscript.